# Structural Performance of Lightweight Aggregate Concrete Reinforced by Glass or Basalt Fiber Reinforced Polymer Bars

**DOI:** 10.3390/polym14112142

**Published:** 2022-05-24

**Authors:** Mohammed A. Abed, Aysha Anagreh, Nikola Tošić, Ola Alkhabbaz, Majd Eddin Alshwaiki, Robert Černý

**Affiliations:** 1John A. Reif, Jr. Department of Civil and Environmental Engineering, New Jersey Institute of Technology, Newark, NJ 07102, USA; 2Department of Structural Engineering, Budapest University of Technology and Economics, Műegyetem rkp. 3, 1111 Budapest, Hungary; ayshayaser.anagreh@edu.bme.hu (A.A.); olaalkhabbaz@outlook.com (O.A.); majdeddinalshwaiki@outlook.com (M.E.A.); 3Civil and Environmental Engineering Department, Universitat Politècnica de Catalunya, Jordi Girona 1–3, 08034 Barcelona, Spain; nikola.tosic@upc.edu; 4Department of Materials Engineering and Chemistry, Faculty of Civil Engineering, Czech Technical University, Thákurova 7, 16629 Prague, Czech Republic; cernyr@fsv.cvut.cz

**Keywords:** lightweight aggregate concrete, expanded clay, glass fiber reinforced polymer, basalt fiber reinforced polymer, shear strength, flexural strength, experimental testing

## Abstract

Lightweight aggregate concrete (LWC) and fiber reinforced polymer (FRP) reinforcement are potentially more sustainable alternatives to traditional steel-reinforced concrete structures, offering several important benefits. To further the knowledge in this area, the physical–mechanical properties of LWC produced with 0%, 50%, and 100% expanded clay aggregate were assessed. Subsequently, the flexural behavior of LWC beams reinforced with steel reinforcement and glass and basalt FRP bars was tested. The results of the experimental program allowed quantifying of the effect of expanded clay aggregate incorporation on LWC properties. The use of FRP reinforcement was also compared to steel-reinforced concrete beam behavior. The results of this study can provide additional support for the use of innovative materials such as LWA and FRP reinforcement.

## 1. Introduction

In traditional concrete production, coarse and fine aggregates are mixed with cement, water, and admixtures to provide the desired mix. By replacing coarse normal weight aggregate with lightweight aggregate (LWA) such as volcanic pumice, clay, oil palm, and fly ash, lightweight aggregate concrete (LWC) is produced. The generic term LWC refers to any concrete with an oven-dry density lower than 1850 kg/m^3^ [1]. In structural terms, LWC is primarily used to reduce the permanent loads (and as a consequence, seismic loads) of structures. However, the use of LWA can offer economic and environmental advantages as well [2,3].

In terms of material, the LWA used to produce the LWC, can be either artificial aggregate, such as expanded shale and expanded clay, or natural aggregates such as scoria and pumice [4,5]. The use of LWA can negatively affect the workability of concrete, due to its porous structure, which rapidly absorbs mixing water. Several studies concluded that using LWA to produce concrete reduces compressive strength.

For example, Hedjazi [6] studied the effect of using different types of LWA on concrete strength and concluded that by increasing the substitution ratio of any type of LWA, compressive strength decreases. As revealed, the interfacial transient zone does not represent a major weakness of LWA concrete, rather the mechanical characteristics of the used LWA play a crucial role. The splitting tensile strength and flexural strength of LWC depend on the compressive strength of the concrete and are thus reduced by using LWA [7,8]; however, the properties of LWC depend on the type of LWA and the curing condition [9]. Ultimate shear strength transferred across the interface of LWC also decreases, as it depends on the LWA type and interface conditions [10]. Hora [11] also indicated that the reduction in LWC strength is due to the high porosity and the high ability of LWA to absorb water, which directly affected the mechanical behavior of concrete.

Over the years, expanded clay—a kind of LWA—has proven to provide good performance to LWC in the long run, due to the close compatibility of stiffness between the LWA and the cement paste, which reduces micro-cracking (which otherwise appears easily in the interfacial transition zone at early hydration due to the high absorption rate [12]). It is used worldwide due to its low production cost, good sound and thermal insulation, low thermal conductivity, and low expansion coefficient when compared with normal weight aggregates [13]. Expanded clay is proven to have high quality, durability, and toughness. Moreover, it is highly recommended as an eco-efficient material for non-dangerous waste materials to be reclaimed; thus, there is no need to dispose of it in a landfill, and this benefits the environment and provides economic advantages [14].

Sonia and Subashini [15] experimentally investigated the effect of expanded clay on the mechanical properties of LWC and found that it decreases both the compressive and splitting tensile strengths. Maghsoudi et al. [16] also observed the same behavior, especially in the case of compressive strength. However, the flexural strength of expanded clay LWC is higher compared with the concrete produced by normal weight aggregates [17]. It should be noted that the particle size of the used LWA has an important role in the strength parameters of a designed LWA concrete in the hardened state, as smaller particles provide higher compressive strength than larger ones. Using expanded clay in self-compacting concrete (SCC) causes more brittle failure [16]. Sonia and Subashini [15] pointed out that expanded clay, if produced well, can be a reason to improve the fresh proprieties of SCC, because of the spherical shape of the aggregate particles.

In addition to innovations at the level of concrete constituents, such as the use of LWA, innovations for structural, durability, and environmental beneficiation of concrete also include reinforcement. In this regard, fiber reinforced polymers (FRPs) are increasingly used as an alternative reinforcing material to steel reinforcement [18]. FRP is a composite material (such as carbon, aramid, steel, basalt, and glass) that is formed by incorporating different fibers into polymeric resins [19,20]. It is highly resistant to corrosion and chemical attack, with a high strength-to-weight ratio [21]. In terms of structural application and design, serviceability criteria are often used as deciding factors in designing FRP-reinforced concrete sections, as FRP generally has a low modulus of elasticity and high tensile strength without a yielding point [22].

Glass fiber reinforcement polymers (GFRPs) and basalt fiber reinforcement polymers (BFRPs) are commonly used and widely applied FRPs in reinforced concrete structures. The use of GFRP increases flexural stiffness, reduces deformability, and increases the resistance capacity of concrete structures. It is an anisotropic composite material with high tensile strength compared with steel bars [23,24].

The deflections and crack width of FRP-reinforced concrete members should be greater than those of steel-reinforced concrete members due to the lower modulus of elasticity and higher tensile strength compared with steel bars [25]. Therefore, the required amount of GFRP to reinforce an LWC member must be larger compared with a steel-reinforced LWC member to achieve the preferable failure mode; otherwise, LWC crushing occurs before the rupture of the GFRP [26].

BFRP has higher tensile strength compared to GFRP and better bonding to the concrete [27]. The increased longitudinal ratio of BFRP enhances the shear strength and stiffness of the concrete beams, but the failure mode becomes more brittle [28]. For instance, Elgabbas et al. [29] observed that the cracking moment of BFRP beams, by applying the bending test until failure, was less than the values predicted by the ACI 440.1R [30] and CAN/CSA-S806 [31] equations. The failure modes of FRP beams are either concrete crushing in the compression zone or rupture of the FRP itself [21]. El Refai and Abed [28] tested the shear strength of BFRP beams without shear reinforcement, and observed that the behavior of BFRP- or GFRP-reinforced beams was quite similar.

Considering the results from the literature presented above, a clear gap was identified in relation to the knowledge of the behavior of FRP reinforcement embedded in LWC. Specifically, the effect of LWA incorporation on mechanical properties needs to be further studied, as well as the structural behavior of FRP-reinforced LWC members. In particular, illuminating the differences between BFRP and GFRP is needed.

Therefore, within this study, an experimental program was conceptualized and executed. For this purpose, the physical–mechanical properties of LWC produced with 0%, 50%, and 100% expanded clay aggregate were assessed. Subsequently, the behavior of LWC beams reinforced with steel reinforcement and GFRP and BFRP bars was tested in three-point bending. The results of the experimental program allowed quantifying of the effect of expanded clay aggregate incorporation on LWC properties.

## 2. Materials and Methods

### 2.1. Materials

Cement, fine aggregate, coarse aggregate, water, and superplasticizer have been used to produce high strength SCC. Hence, a relatively low *w*/*c* ratio of 0.35 and a relatively high cement content of 500 kg/m^3^ were selected. The cement was Ordinary Portland Cement (OPC) CEM I 42.5 N tested per BS EN 196-2 [32]; its chemical and physical characteristics can be seen in Table 1.

Local natural quartz river (Danube River in Hungary) sand and two types of coarse aggregate were used: natural aggregate (NA) and expanded clay with maximum sizes of 8 mm. The choice for maximum aggregate size both for the NA and expanded clay was justified from the point of view of SCC production, which requires smaller maximum aggregate sizes. Fine and coarse aggregate were used in mass proportions of 45% and 55%, respectively.

Expanded clay has been used as LWA with different proportions (0%, 50%, and 100%) of normal coarse aggregate mass. The sieve analyses for the three blends of aggregate are shown in Figure 1. Expanded clay is shown in Figure 2, while its physical properties are presented in Table 2. To achieve the required strength and workability, as well as to satisfy the European guidelines for SCC, EFNARC (2005), both BASF Glenium C300 and BASF Glenium 51 were used.

Sand dune coated GFRP, BFRP, and steel B500B bars with diameters of 10, 14, and 8 mm, respectively, were used as tension reinforcement in the tested beams. The main properties of the used reinforcements are shown in Table 3.

### 2.2. Mix Design

The mixing procedure was partitioned into three stages: in the first stage the coarse and fine aggregate and cement were added and mixed; in the second stage, water was added; and finally, both BASF Glenium C300 and BASF Glenium 51 superplasticizers were added. This procedure allowed achievement of the required workability and homogeneity of the mixes. Three mixtures were produced with an increasing percentage of LWA replacing NA: LW0% is the mix with 0% of expanded clay, LW50% is the mix with 50% of expanded clay and 50% of NA, and LW100% is the mix with 100% expanded clay. All mix proportions are presented in Table 4.

### 2.3. Test Setups and Specimen Preparation

In order to ensure proper SCC behavior, slump flow and V-funnel tests were conducted directly after mixing, according to the European guidelines for SCC, EFNARC [33] (see Figure 3). The slump flow test was used to assess the workability and the consistency of the concrete, while the V-funnel test was used to measure the flowability of the mixture under its self-weight. Then, the fresh density was measured, and the samples were cast in steel molds of different sizes to obtain the standard specimens for each test. Two mixtures were made for each test, where the first mixture had 100% of normal coarse aggregate and 0% expanded clay aggregate, and the second mixture had 100% of expanded clay aggregate with 0% coarse aggregate.

Cubic 150 mm × 150 mm × 150 mm specimens were used for compressive strength tests, 70 × 70 × 250 mm prisms were used for flexural strength tests, and Ø150 mm × 300 mm cylinders were used for the splitting tensile strength and shear strength tests. For every single property, three specimens were tested, and the average was recorded. All specimens were kept in their molds for 24 h at a temperature of 22 ± 1 °C, after which they were de-molded and immersed in water for 7 days, and then kept under laboratory conditions at 20 ± 2 °C and 35% relative humidity, until the testing date at the age of 28 days.

Compressive strength, splitting tensile strength, and three-point bending tests were conducted in accordance with BS EN 12390-3 [34], CEN EN 12697-23 [35], and BS EN 12390-5 [36], respectively. For testing the compressive strength, three 150 mm cubes were tested, while three cylinders with a diameter of 150 mm and a height of 300 mm were tested to evaluate the splitting tensile strength. For the three-point bending test, three prisms with dimensions of 70 mm × 70 mm × 250 mm were tested for each type of concrete. However, shear strength was tested using a novel experimental method, consisting of a notched cylindrical push-off specimen, with a diameter of 150 mm and a height of 300 mm, which creates two stress-free zones. New boundary conditions were created to transform the compression stress into shear in a limited area. The S-shape of such a specimen allows for a longitudinal slip, which ensures the occurrence of shear stress in a plane by loading with two forces in equilibrium with each other, without the need for applying additional forces on the boundary to ensure equilibrium. The specimen was notched in two symmetrical areas with a 10 mm thick, 75 mm deep notch, and with the spacing between the notch and base of 100 mm on both sides. After measuring the failure load (*P*) for each test, the compressive, splitting tensile, flexural, and shear strengths were calculated in MPa using the equations presented in Table 5.

Finally, twelve reinforced concrete beams were tested: four of them reinforced with GFRP, four of them reinforced with BFRP, and the last four reinforced with steel bars. For each reinforcement type, two beams were tested: two from concrete LW0% and two from concrete LW100%. The adopted nomenclature for the beams was the following: St-LW0% and St-LW100% for the steel-reinforced beams with 0% and 100% of LWA, respectively. Gl-LW0% and Gl-LW100% for the GFRP beams with 0% and 100% of LWA, respectively, and finally Ba-LW0% and Ba-LW100% for the BFRP beams with 0% and 100% of LWA, respectively.

The beams had dimensions of width/height/length = 100/150/1100 mm. The beams were reinforced with two bars as bottom longitudinal reinforcement, with a clear concrete cover of 30 mm, while no shear reinforcement was used. Table 6 shows the details of the test beams including reinforcement ratio (*ρ_f_*), balanced reinforcement ratio (*ρ_fb_*), reinforcement types and numbers, beams nomenclature, and concrete type.
(1)$$\rho_{f}=\frac{A_{f}}{bd}$$
(2)$$\rho_{fb}=0.85\beta_{1}\frac{f\prime_{c}}{f_{fu}}\frac{E_{f}\varepsilon_{cu}}{E_{f\;}\varepsilon_{cu}+f_{fu}}$$
where:*β*_1_—0.65 (for high strength concrete).*f_fu_*—design tensile strength of the reinforcement bars (MPa).*E_f_*—modulus of elasticity of the reinforcement bars (MPa).*ε_cu_*—ultimate strain of the concrete.*f_c_^’^*—specified compressive strength of the concrete.*b*—width of the beam.*d*—distance from extreme fiber in compression to the center of reinforcement (mm).*A_f_*—area of longitudinal reinforcement (mm^2^).

The beams were simply supported with a clear span of 900 mm and loaded in three-point bending using a hydraulic jack located at the center of the beam, allowing for the beams to deflect under monotonic load. A 600 kN hydraulic actuator anchored to an independent steel frame was used to apply a monotonic increasing load at the mid-span of the beam. The beams were tested under displacement-controlled loading at the rate of 1 mm/min, until failure of the beam by reinforcement rupture or concrete crushing. The mid-span deflections were measured using linear variable differential transformers (LVDTs). Two LVDTs were used to measure and record the vertical deflection at the mid-span. A load cell and the LVDTs were connected to a data logger to obtain vertical deflection readings and applied load. The beams’ dimensions and reinforcement are shown in Figure 4.

## 3. Results and Discussion

### 3.1. Physical–Mechanical Properties of LWC

The results of physical–mechanical property tests are given in Table 7. The results point to the effect of LWA on several properties of LWC. As the LWA dosage increases, the dose of superplasticizer required to compensate for the loss of workability of the LWC increases. This behavior is related to the high absorption capacity of LWA. However, the LWC mixtures meet the European guidelines of SCC [33], with a classification of SF1 for slump flow and VF1 for the flow time. Moreover, all the mixtures meet the workability process window suggested for high strength SCC [37]. The effect of LWA is clearly seen in the reduction in density by 13.5% and 21.3% for 50% and 100% of LWA, respectively. Thus, the concretes LW50% and LW100% could be classified as Density Classes 2.0 and 1.8, respectively, according to Eurocode 2 [38].

Similar to workability, the mechanical properties of LWC also exhibit a decrease with an increasing percentage of LWA. In terms of compressive strength, adding 50% and 100% of LWA decreased compressive strength by 23% and 28%, respectively. This reduction in compressive strength is most likely due to the high porosity of the expanded clay LWA and its consequent low density and crushing strength compared to NA. Applying the Eurocode 2 classification, the concretes LW0%, LW50%, and LW100% could be classified as C70/77, LC50/55, and LC45/50, respectively.

As for splitting tensile strength, the reduction was 17% and 25% for 50% and 100% of LWA substitution, respectively, whereas, in the case of flexural strength, the decrease was 10% and 20% for 50% and 100% of LWA substitution, respectively. The decreases in tensile strength are slightly smaller than those for compressive strength since aggregate crushing strength is not the determining parameter, but rather their tensile strength and the cement paste–aggregate bond. However, due to the higher porosity and water absorption of LWA, the pore area and thickness of the interfacial zone are increased in LWC, preventing cement particles from binding tightly to the expanded clay LWA [6,17,39].

Finally, in terms of shear strength, the reduction was 20% and 34% for LW50% and LW100%, respectively, relative to LW0%. The greater brittleness of LWA has been previously postulated as an explanation for the reduction in shear strength of LWC compared with normal weight concrete [40]. Nonetheless, it should be noted that the effect is not linear and the obtained values demonstrate the suitability of both LWCs for structural applications.

### 3.2. Behavior of Reinforced LWC Beams

In order to investigate and compare the flexural behaviors of 0%LWA and 100%LWA beams reinforced with steel, GFRP, and BFRP bars, a three-point bending test was applied on two beams of each concrete type with each reinforcement type. A load cell and two LVDTs were placed at the mid-span and connected to the data logger to get the vertical deflection values. In accordance, the LVDTs were connected to an automatic data-acquisition system, which was connected to a computer to record the loading, deflections, and strains in the reinforcement and the surrounding concrete (Figure 5).

### 3.3. Load–Displacement Relationships

Load–displacement graphs were generated on the computer while performing the three-point bending test over a clear span of 900 mm. Furthermore, crack propagation and failure type of the specimens were observed. The load–displacement relationships of St-0%LWA and St-100%LWA beams are shown in Figure 6. St-0%LWA beams behaved as expected, where first they acted linearly until they reached the yielding point and then they behaved plastically until the maximum load–displacement was reached. The maximum load of St-0%LWA beams was 41.45 kN and the corresponding mid-span deflection was 13.49 mm. After reaching the maximum load–displacement points, the cracks propagated rapidly and the beam collapsed at a load of approximately 25 kN, with a displacement of 25 mm. However, the St-100%LWA beam behaved similarly to the normal weight concrete beams in terms of reaching the yielding load of around 32 kN with a mid-span deflection of 6 mm. However, the maximum load of 35.29 kN was reached at a displacement of only 9.36 mm, indicating a much more brittle and less ductile behavior than in the case of St-0%LWA beams, due to the weaker aggregate causing crushing in the compression zone. It should be noted that only one specimen was tested in this case, limiting the possibility of drawing more definite conclusions.

Both Gf-0%LWA and Gf-100%LWA faced a sudden concrete crushing failure after reaching the maximum load (Figure 7). This behavior is in accordance with the results of Junaid et al. [41]. In the case of Gf-0%LWA beams, the first cracking load was approximately 8 kN and the corresponding displacement was around 3 mm, however, the maximum reached load at failure was 31.98 kN with a mid-span deflection of 7.59 mm. The initial cracking load of Gf-100%LWA beams was the same as Gf-0%LWA beams, where the first cracking load was around 8 kN with a displacement of around 3 mm. The failure load of Gf-100%LWA beams was 26.58 kN with a mid-span deflection of 7.51 mm.

Bf-0%LWA and Bf-100%LWA beams behaved similarly to the specimens reinforced with GFRP bars, where the load and the displacement increased linearly until reaching the failure point, and hints the studied beams collapsed, as is observed in Figure 8, which presents the load (kN)–displacement (mm) relationship of Bf-0%LWA and Bf-100%LWA. Regardless of the concrete type, the first crack load was around 5 kN. The failure load of Bf-0%LWA beams was 35.45 kN, with a mid-span deflection at a failure of 7.89 mm. Bf-100%LWA beams did not face a sudden failure as in the case of Bf-0%LWA beams, and this behavior is in accordance with the results of Urbanski et al. [42]—these beams reached failure at a load of 25.2 kN and a displacement of 7.26 mm.

In agreement with the results of Akbarzadeh Bengar et al. (2021) [43], the studied beams reinforced with steel bars, St-0%LWA and St-100%LWA, had higher mid-span deflections at failure by 38% and by 22% than those of the Gf-0%LWA and Gf-100%LWA beams, respectively. Moreover, the first cracking load of the steel-reinforced beams occurred at higher loads than those of the GFRP beams; thus, the pre-cracking and post cracking of the steel-reinforced beams are higher than those of beams reinforced with GFRP. Table 8 presents the maximum deflection at the maximum load, while Figure 9 presents the effect of the type of reinforcing bar on the load–displacement relationship of normal weight concrete and LWC. The lower stiffness of the FRP beams can also be seen from the figure.

Compared to steel reinforced beams, BFRP beams are lower in post crack stiffness, due to the lower modulus of elasticity of BFRP bars The pre-posting stiffness of steel reinforced beams is higher compared to BFRP beams or GFRP beams, where the first cracking loads of St-0%LWA and St-100%LWA are higher than Bf-0%LWA and Bf-100%LWA by 29% and 44%, respectively. It is also worth mentioning that the first cracking load of the GFRP beams is higher than that of BFRP beams by around 38%.

Although the FRP-reinforced beams exhibited more abrupt and sudden failures, they still reached considerable loads, relative to the steel-reinforced beams, as well as sufficiently high deflections (in the order of *L*/100–*L*/120) for enabling detection prior to failure. As such, they can be considered suitable for structural use.

### 3.4. Cracking Moment (M_cr_)

The experimental results of the cracking moment of steel-reinforced beams are higher compared to any of the fiber reinforced polymer beams, and that is due to their higher flexural capacity and the modulus of elasticity of steel bars. Table 9 presents the cracking moment (*M_cr_*) calculated using the ACI 440.1R [30] expression and the experimental values of the tested beams.
(3)$$M_{cr}=\frac{0.62\;\lambda \sqrt{f\prime_{c}}\;I_{g}}{Y_{t}}\;\;$$
where:*M_cr_*—first cracking moment.*λ*—(= 0.75) modification factor reflecting the reduced mechanical properties of LWC, and *λ* = 1.0 for normal weight concrete.*I_g_—*moment of inertia of the gross section.*Y_t_—*distance from the centroid axis of the gross section, neglecting reinforcement, to the tension face.

For all the beams, it shows that LWA decreases the *M_cr_* of the beams, where St-0%LWA, Gf-0%LWA, and Bf-0%LWA have a higher *M_cr_* by 26%, 24%, and 35% compared with St-100%LWA, Gf-100%LWA, and Bf-100%LWA, respectively. The *M_cr_* predicted from the ACI code equation is lower or equal to the experimental values, where the theoretical *M_cr_* values of St-0%LWA and Gf-0%LWA are lower by 31% and 1% compared to the experimental values. In accordance, the theoretical *M_cr_* of St-100%LWA and Bf-100%LWA are 41% and 17% lower compared with the experimental *M_cr_* values.

### 3.5. Crack’s Pattern and Width

While applying the three-point bending test on the reinforced beams, vertical cracks started to occur at the bottom surface of the tested beams, especially in the case of normal weight concrete beams, as they occurred initially due to the early shrinkage of the tested specimens. St-0%LWA developed flexural cracks due to the tensile stresses, while St-100%LWA faced diagonal shear cracks with an angle of around 45°, thereby, with load increment, the shear cracks reached to the point load and failure occurred. GFRP beams, in both cases, faced a combination of flexural and shear cracks patterns with load increment, due to shear and tensile stresses. For BFRP beams, flexural cracks started to form in the bottom of the beam, followed by the occurrence of shear cracks, where they propagated toward the loading point. At the final stage of loading, crack width increased, and the cracks propagated towards the compression zone, reached the point load, and thus the reinforced beams failed.

The inclinations of the critical shear cracks of Bf-100%LWA were in the range of 43–48°, while they were in the range of 56–66° for normal weight beams. The beams failed in diagonal shear failure due to the absence of shear reinforcement, and since the transverse reinforcement was not applied, therefore, the development of inclined cracks caused the shear and brittle failure. In accordance, horizontal cracks were observed along with the longitudinal tensile reinforcement of the BFRP beams due to the absence of the anchorage reinforcement, but their occurrence did not affect the failure mode. These cracks are generated due to the high deformation of the BFRP, causing a slippage between the reinforcement bars and the surrounding concrete. Sketches for the cracking patterns at failure for all the beams are shown in Figure 10. Importantly, no direct relationship could be established between the bar or concrete type and the failure mode. Steel-reinforced beams showed higher crack width values compared with GFRP beams and BFRP beams. Furthermore, LWC beams developed lower crack widths than those of normal weight concrete, where the average crack width of St-0%LWA and Gf-0%LWA was 1.9 mm and 0.6 mm, respectively, while they were 1.7 mm and 0.1 mm for St-100%LWA and Gf-100%LWA, respectively. In accordance, the crack widths of St-100%LWA and Bf-100%LWA were lower by 11% and 33% than those of St-0%LWA and Bf-0%LWA, respectively. Table 10 presents the ultimate crack width of all the beams.

## 4. Conclusions

In this study, the combined use of expanded clay LWA and FRP reinforcement was investigated and the effects of these materials on the physical–mechanical properties of LWC and its structural behavior were investigated. Based on the obtained experimental results, the following conclusions can be reached:Fresh- and hardened-state physical and mechanical properties of SCC decreased by increasing the dose of expanded clay LWA.By using GFRP or BFRP bars, the maximum load and maximum deflection of beams are reduced compared to the beams reinforced with steel bars. However, in LWC, the effect of FRPs on the maximum deflection is smaller compared with the effect in normal weight concrete or LWC beams reinforced with steel bars.For LWC, the beams reinforced by GFRP or BFRP perform similarly in terms of maximum load, maximum deflection, crack width, and cracking moment. The maximum deflection of the beams reinforced by steel bars is decreased when LWC is used.

The results of this study can provide additional support for the use of innovative materials such as LWA and FRP reinforcement. Further studies should be carried out to investigate other aspects of both material and structural behavior, in particular bond characterization tests, to further deepen the knowledge and understanding of LWC reinforced with innovative reinforcements.

## Figures and Tables

**Figure 1 polymers-14-02142-f001:**
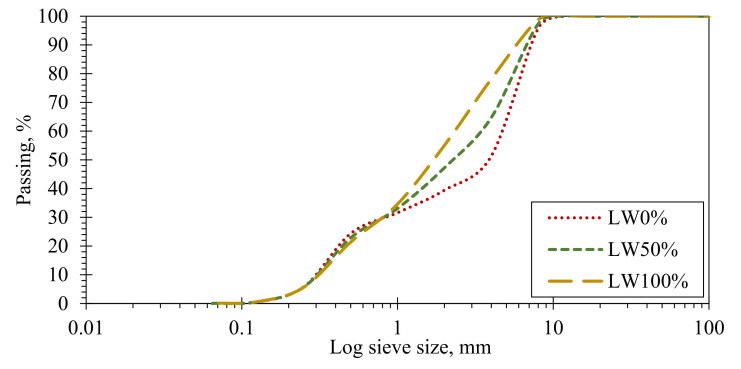
Sieve analysis for produced aggregate blends.

**Figure 2 polymers-14-02142-f002:**
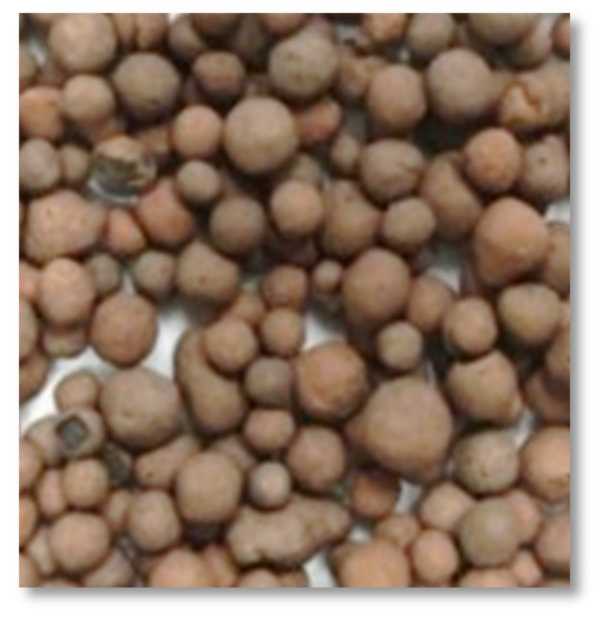
Expanded clay.

**Figure 3 polymers-14-02142-f003:**
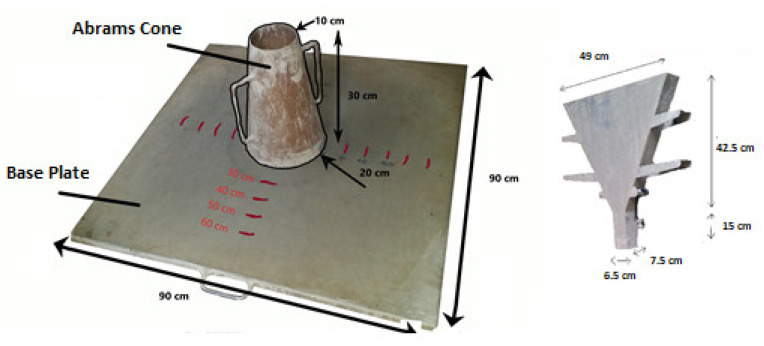
Equipment for slump flow and V-funnel tests.

**Figure 4 polymers-14-02142-f004:**
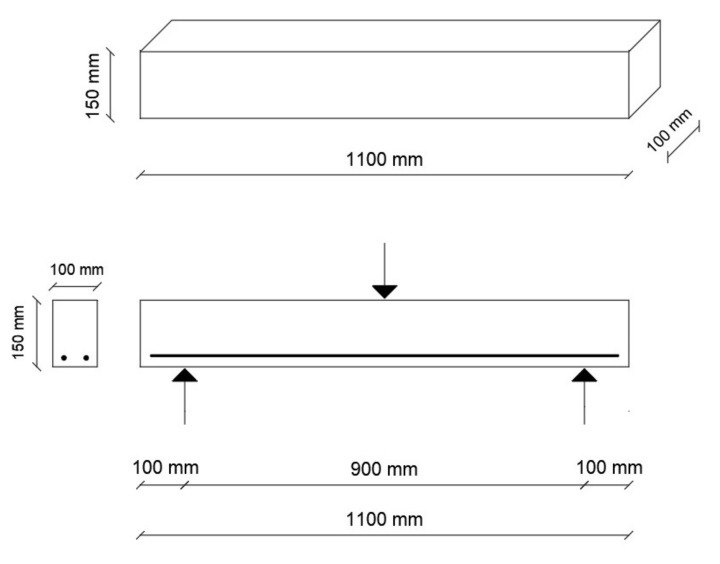
Beam dimensions and reinforcement layout.

**Figure 5 polymers-14-02142-f005:**
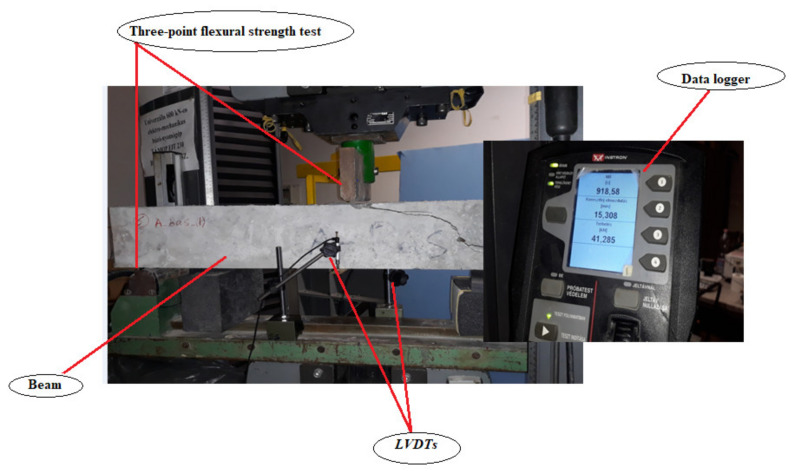
Three-point bending strength test and data logger.

**Figure 6 polymers-14-02142-f006:**
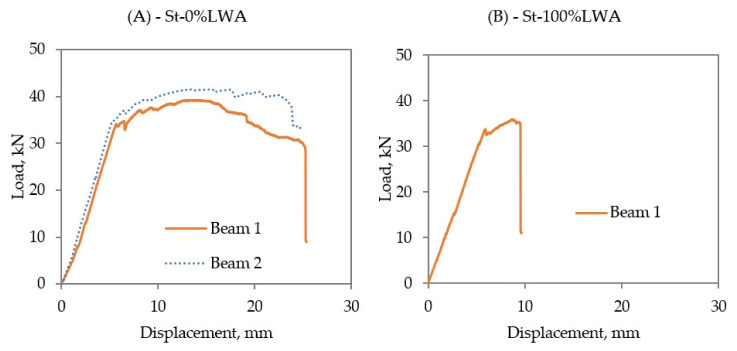
Load–displacement relationship of (**A**) St-0%LWA, and (**B**) St-100%LWA beams.

**Figure 7 polymers-14-02142-f007:**
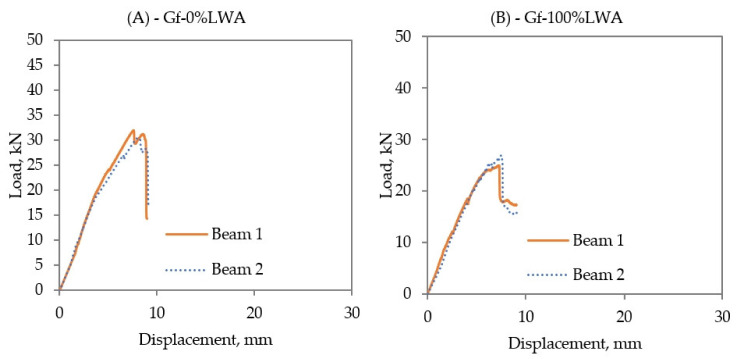
Load–displacement relationship of (**A**) Gf-0%LWA, and (**B**) Gf-100%LWA beams.

**Figure 8 polymers-14-02142-f008:**
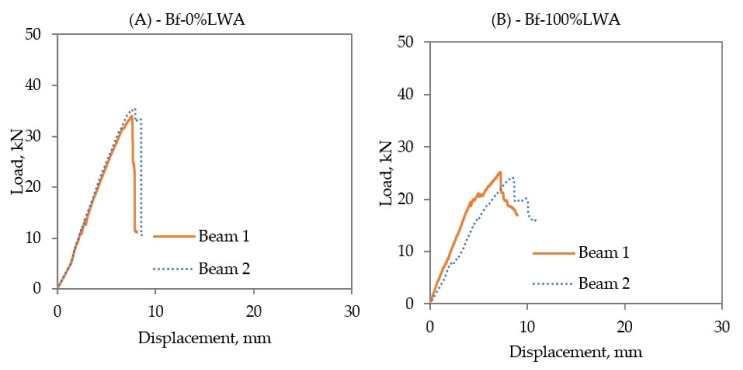
Load–displacement relationship of (**A**) Bf-0%LWA, and (**B**) Bf-100%LWA beams.

**Figure 9 polymers-14-02142-f009:**
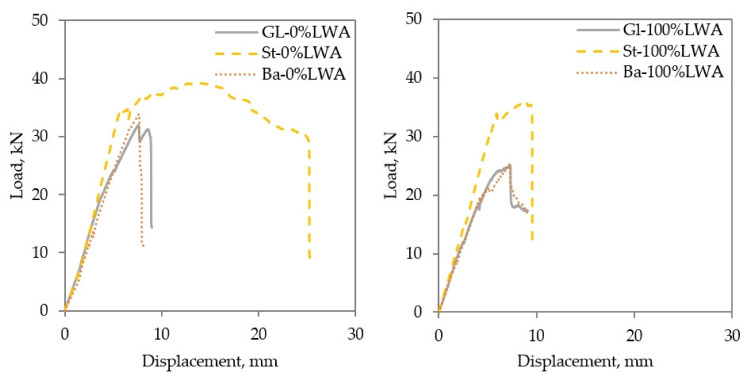
Effect of type of reinforcing bar on the load–displacement relationship of (**left**) normal weight concrete and (**right**) LWC.

**Figure 10 polymers-14-02142-f010:**
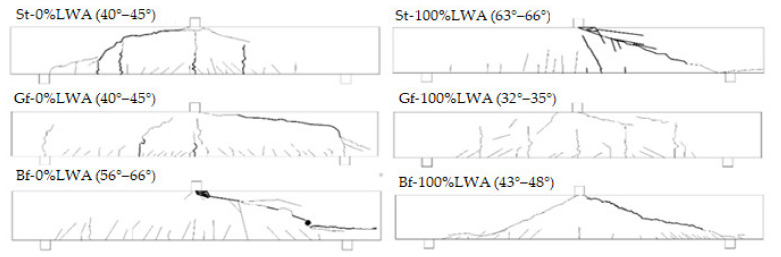
Sketches for cracking patterns at failure for all beams.

**Table 1 polymers-14-02142-t001:** Chemical compositions and physical properties of cement.

Measured Property of CEM I												
Density (g/cm^3^)	Specific Surface Area (cm^2^/g)	Loss on Ignition (%)	Chloride (%)	Free CaO (%)	SiO_2_ (%)	CaO (%)	MgO (%)	Fe_2_O_3_ (%)	Al_2_O_3_ (%)	SO_3_ (%)	Na_2_O (%)	K_2_O (%)
3.02	3326	3	0.04	0.71	19.33	63.43	1.45	3.42	4.67	2.6	0.33	0.78

**Table 2 polymers-14-02142-t002:** Properties of LWA (expanded clay).

Property	Expanded Clay
Oven-dry density (kg/m^3^)	2620
Bulk density (kg/m^3^)	359
Particle density (kg/m^3^)	650
Water absorption (%)	18.3
Particle porosity (%)	75.2
Crushing resistance (MPa)	2

**Table 3 polymers-14-02142-t003:** Properties of reinforcement bars.

Property	Tensile Strength (MPa)	Yield Strength (MPa)	Modulus of Elasticity (MPa)	Ultimate Strain (%)	Density (kg/m^3^)	Diameter (mm)	-
Steel B500B	540	500	200,000	5	7850	8	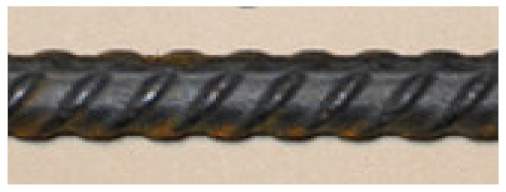
GFRP	920	-	55,500	1.68	2100	10	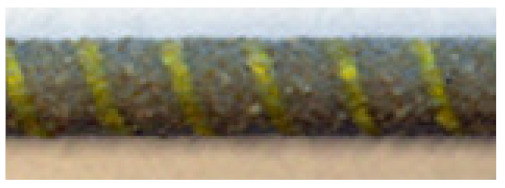
BFRP	1100	-	70,000	2.2	1900	14	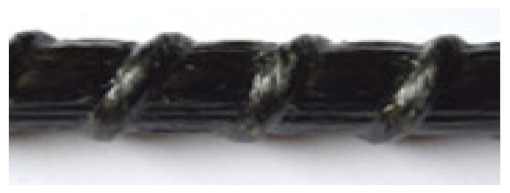

**Table 4 polymers-14-02142-t004:** Concrete mix proportions.

Mix	Proportions in kg/m^3^						
	Cement	Fine Aggregate	Coarse Aggregate		Glenium C300	Glenium 51	Water
		Natural Sand	NA	LWA			
		0/4 (mm)	4/8 (mm)				
LW0%	500	785	960	–	0.75	0.75	175
LW50%	500	785	480	232	1.50	2.00	175
LW100%	500	785	–	464	2.25	3.75	175

**Table 5 polymers-14-02142-t005:** Formulas for calculating the mechanical strengths.

Test	Formula	Notations
Compressive strength	$\frac{P\;}{A_{c}\;}$	*P* = load at failure (kN) *A_c_* = compression area (mm^2^) *L_T_* = length of the cylinder (mm) *D* = diameter of the cylinder (mm) *b* = width of the prism (mm) *d* = depth of the prism (mm) *L_F_* = length between supports of the prism (mm) *A_shear_* = shear area (mm^2^)
Splitting tensile strength	$\frac{2P\;}{\pi DL_{T}\;}$	
Bending strength	$\frac{3PL_{F}}{2bd^{2}}$	
Shear strength	$\frac{P\;}{A_{shear}\;}$	

**Table 6 polymers-14-02142-t006:** Details of test beams.

Concrete Type	Beam Nomenclature	Beam No.	Reinforcing Material	Compressive Strength (MPa)	Span/Depth	Longitudinal Reinforcement			
						Amount of Reinforcement	*ρ_f_* (%)	*ρ_fb_* (%)	*ρ_f_*/*ρ_fb_*
Normal weight concrete	St-0%LWA	Beam 1	Steel	78	7.76	2 No. 8	0.87	3.21	0.27
		Beam 2							
	Bf-0%LWA	Beam 1	BFRP		7.96	2 No. 14	2.72	1.36	2.00
		Beam 2							
	Gf-0%LWA	Beam 1	GFRP		7.83	2 No. 10	1.37	0.70	1.95
		Beam 2							
Lightweight concrete	St-100%LWA	Beam 1	Steel	56	7.76	2 No. 8	0.87	2.31	0.37
		Beam 2							
	Bf-100%LWA	Beam 1	BFRP		7.96	2 No. 14	2.72	0.98	2.78
		Beam 2							
	Gf-100%LWA	Beam 1	GFRP		7.83	2 No. 10	1.37	0.52	2.63
		Beam 2							

**Table 7 polymers-14-02142-t007:** Physical–mechanical properties of the tested concretes.

Mix	Slump Flow (mm)	V-Funnel (s)	*ρ* (kg/m^3^)	*f_c_* (MPa)	*f_ct,sp_* (MPa)	*f_ct,fl_* (MPa)	*τ_c_* (MPa)
LW0%	670	7.0	2349	78.0	4.0	6.2	10.3
LW50%	635	6.7	2030	60.0	3.3	5.6	8.3
LW100%	620	6.0	1851	56.0	3.0	5.0	6.8

**Table 8 polymers-14-02142-t008:** Average values of the maximum deflection at the maximum load.

Property	St-0%LWA	St-100%LWA	Bf-0%LWA	Bf-100%LWA	Gf-0%LWA	Gf-100%LWA
Max. load (kN)	41.45	35.29	35.45	25.20	31.98	26.58
Max. deflection (mm)	13.49	9.36	7.86	7.26	7.59	7.51

**Table 9 polymers-14-02142-t009:** First moment of crack calculated by ACI equation and experimental values.

Beam	Experimental *M_cr_* (kNm)	Calculated *M_cr_* (kNm)	Mode of Failure
St-0%LWA	2.99	2.05	Reinforcement yielding and Concrete crushing
St-100%LWA	2.22	1.30	Reinforcement yielding and Concrete crushing
Gf-0%LWA	2.07	2.05	Concrete crushing
Gf-100%LWA	1.57	1.30	Concrete crushing
Bf-0%LWA	2.05	2.05	Concrete crushing
Bf-100%LWA	1.33	1.30	Concrete crushing

**Table 10 polymers-14-02142-t010:** Ultimate crack width of all beams.

Property	St-0%LWA	St-100%LWA	Gf-0%LWA	Gf-100%LWA	Bf-0%LWA	Bf-100%LWA
Crack width at ultimate load (mm)	1.9	1.7	0.6	0.1	0.3	0.2

## Data Availability

The data presented in this study are available on request from the corresponding author.

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
