# Peer review of "Structural Performance of Lightweight Aggregate Concrete Reinforced by Glass or Basalt Fiber Reinforced Polymer Bars"

_polymers, 2022, doi:10.3390/polym14112142_

Round 1
Reviewer 1 Report
This paper presents an experimental study of LWC beams reinforced with FRP bars. The topic is within scope of this journal and may be interested by the readers. The authors may consider the following comments:
- It is not true that FRP reinforced RC beams have a higher yied strain than the steel reinforced RC beams. The author should correct the statement is the introduction section "the deflections and crack width of FRP-reinforced concrete members should be greater than that of steel reinforced concrete members due to the lower modulus of elasticity and higher yield strain compared with steel bars". Beams with FRP reinforcement do not have a 'yield strength'.
- BFRP has higher tensile strength compared with GFRP and better bonding to the concrete: some references should be cited to demonstrate this statement.
- The author mentioned in the introduction section that a clear gap was identified in relation to knowledge of the behavior of FRP reinforcement embedded in LWC. Based on the study presented in the paper, the behavior of FRP reinforcement in terms of strain development in FRP and bonding performance should be illustrated in the test result section.
- Recent studies on FRP reinforced concrete structures (e.g., J. Build. Eng. 47 (2022) 103782; Compos. Struct. 285 (2022) 115143; Compos. Struct. 285 (2022) 115268) should be cited to widen the insights of the introduction.
- As FRP may lead to a smaller stiffness of the beams, comparisons of stiffness of the beams are required.
- The information about the reinforcement details of the test beams is missing.
- The authors should explain why three different diameters of steel, GFRP and BFRP bars were adopted.
- Please explain why the maximum deflection of the beams reinforced by steel bars is decreased when LWC is used.
- The conclusion that the SCC beam decreased by increasing the dose of LWA due to the high porosity is not sound. The porosity is not reported in this paper.
Author Response
Response to Reviewers’ comments and description of changes in the revised Manuscript ID sustainability-1702429, submitted to Polymers
General comments:
Authors’ general comments to Editor and Reviewer:
We are grateful to the Editor and Reviewers for the time dedicated to revising our paper and their comments. We have done our best to implement all suggested changes to the manuscript and we are certain this has helped improve its quality. Our responses to the review comments are below, and changes that we have made to the paper are highlighted in the revised manuscript submission.
Reviewer #1:
This paper presents an experimental study of LWC beams reinforced with FRP bars. The topic is within scope of this journal and may be interested by the readers. The authors may consider the following comments:
- It is not true that FRP reinforced RC beams have a higher yied strain than the steel reinforced RC beams. The author should correct the statement is the introduction section "the deflections and crack width of FRP-reinforced concrete members should be greater than that of steel reinforced concrete members due to the lower modulus of elasticity and higher yield strain compared with steel bars". Beams with FRP reinforcement do not have a 'yield strength'.
|
Response: |
Corrected |
|
Notes/actions: |
Page 2, line 84 |
- BFRP has higher tensile strength compared with GFRP and better bonding to the concrete: some references should be cited to demonstrate this statement.
|
Response: |
Reference added |
|
Notes/actions: |
Page 2, line 90 |
- The author mentioned in the introduction section that a clear gap was identified in relation to knowledge of the behavior of FRP reinforcement embedded in LWC. Based on the study presented in the paper, the behavior of FRP reinforcement in terms of strain development in FRP and bonding performance should be illustrated in the test result section.
|
Response: |
Please note that certain observations regarding bond have been written on page 13, lines 372–374. A note on future studies has been added to the Conclusions. |
|
Notes/actions: |
Page 14, lines 406–407. |
- Recent studies on FRP reinforced concrete structures (e.g., J. Build. Eng. 47 (2022) 103782; Compos. Struct. 285 (2022) 115143; Struct. 285 (2022) 115268) should be cited to widen the insights of the introduction.
|
Response: |
Added |
|
Notes/actions: |
Ref. # 18, 21, 25 |
- As FRP may lead to a smaller stiffness of the beams, comparisons of stiffness of the beams are required.
|
Response: |
Corrected |
|
Notes/actions: |
Page 11, lines 313–314 |
- The information about the reinforcement details of the test beams is missing.
|
Response: |
Please see the description on 7, lines 194–198, Tables 3 and 6 and Figure 4. |
|
Notes/actions: |
None |
- The authors should explain why three different diameters of steel, GFRP and BFRP bars were adopted.
|
Response: |
This was set in the scope of the study because the cross-section of the beam is relatively small as 8 mm of steel bar would be enough, the 8 mm of FRP will not provide a comparative maximum load and maximum deflection to steel reinforcing beams. In fact, considering the completely different mechanical properties of used steel alternatives, significant attention was paid rather than the preservation of the mechanical performance. Moreover, the used diameters are based on performed preliminary tests, the literature survey, and material availability. |
|
Notes/actions: |
- |
- Please explain why the maximum deflection of the beams reinforced by steel bars is decreased when LWC is used.
|
Response: |
Corrected |
|
Notes/actions: |
Page 10, line 280 |
- The conclusion that the SCC beam decreased by increasing the dose of LWA due to the high porosity is not sound. The porosity is not reported in this paper.
|
Response: |
Corrected |
|
Notes/actions: |
Page 14, lines 393–394 |
Reviewer 2 Report
The authors submitted an interesting study that is very comprehensive and properly done. I have only a few minor notes that are listed as follows:
- Line 37, sentence: “The use of LWA can negatively affect the workability of concrete and reduce workability, due to…” describes the same thing twice.
- Paragraph between lines 40 and 48 describes mechanical properties reduction of LWC as a result of LWA application. I miss there at least a brief mention about scope of such reduction. The same applies to the paragraph between lines 59 and 66.
- Line 51: there is missing “)” character.
- I recommend to describe what the SCC abbreviation means (although it is widely known).
- Figure 2 is of very poor quality.
- Explain why diameters of 8, 10, and 14 mm of steel, GFRP, and BFRP were chosen, as these significantly differ to each other in terms of load-bearing capacity.
- Flexure strength test of St-100%LWA: It is a bit questionable to make conclusion from only one specimen tested.
Author Response
Response to Reviewers’ comments and description of changes in the revised Manuscript ID sustainability-1702429, submitted to Polymers
General comments:
Authors’ general comments to Editor and Reviewer:
We are grateful to the Editor and Reviewers for the time dedicated to revising our paper and their comments. We have done our best to implement all suggested changes to the manuscript and we are certain this has helped improve its quality. Our responses to the review comments are below, and changes that we have made to the paper are highlighted in the revised manuscript submission.
Reviewer #2:
The authors submitted an interesting study that is very comprehensive and properly done. I have only a few minor notes that are listed as follows:
- Line 37, sentence: “The use of LWA can negatively affect the workability of concrete and reduce workability, due to…” describes the same thing twice.
|
Response: |
Corrected |
|
Notes/actions: |
Page 1, line 37 |
- Paragraph between lines 40 and 48 describes mechanical properties reduction of LWC as a result of LWA application. I miss there at least a brief mention about scope of such reduction. The same applies to the paragraph between lines 59 and 66.
|
Response: |
Modefied |
|
Notes/actions: |
Page 1, lines 43 – 44; page 2, lines 66 – 68. |
- Line 51: there is missing “)” character.
|
Response: |
Corrected |
|
Notes/actions: |
Page 2, line 52 |
- I recommend to describe what the SCC abbreviation means (although it is widely known).
|
Response: |
Corrected |
|
Notes/actions: |
Page 2, line 64 |
- Figure 2 is of very poor quality.
|
Response: |
Modified |
|
Notes/actions: |
Figure 2 |
- Explain why diameters of 8, 10, and 14 mm of steel, GFRP, and BFRP were chosen, as these significantly differ to each other in terms of load-bearing capacity.
|
Response: |
This was set in the scope of the study because the cross-section of the beam is relatively small as 8mm of steel bar would be enough, the 8 mm of FRP will not provide a comparative maximum load and maximum deflection to steel reinforcing beams. . In fact, considering the completely different mechanical properties of used steel alternatives, significant attention was paid rather than the preservation of the mechanical performance. Moreover, the used diameters are based on performed preliminary tests, the literature survey, and material availability. |
|
Notes/actions: |
- |
- Flexure strength test of St-100%LWA: It is a bit questionable to make conclusion from only one specimen tested.
|
Response: |
Corrected |
|
Notes/actions: |
Page 10, lines 281–282 |
Reviewer 3 Report
The paper proposes a study on the Structural Performance of Lightweight Aggregate Concrete Reinforced by Glass or Basalt Fiber Reinforced Polymer Bars. The paper is clear and concise and it addresses an interesting and relevant problem. The English is acceptable, tables are well organized. The experimental tests developed in the manuscript are a good contribution to the field, but the Authors should explain some of their choices. Finally, I suggest a minor revision. In what follows, I list some comments and suggestions that can be addressed by the authors while finalizing the manuscript in a minor revision process.
Lines 74-81: The authors are asked to introduce that there are also SRPs, generlamente they are used as external reinforcement of structural elements. We segger to consider the work: https://doi.org/10.1016/j.engstruct.2022.114084
Line 140: What standard was used to evaluate the parameters shown in Table 3?
Figure 3 is poorly readable.
Lines 216-219: I am a little confused, was the test conducted in displacement control with a rate 1 mm/min or in load control "apply a monotonic increasing load"? The authors are invited to rewrite the sentence.
Line 393: The authors are invited to expand the conclusions.
Author Response
Response to Reviewers’ comments and description of changes in the revised Manuscript ID sustainability-1702429, submitted to Polymers
General comments:
Authors’ general comments to Editor and Reviewer:
We are grateful to the Editor and Reviewers for the time dedicated to revising our paper and their comments. We have done our best to implement all suggested changes to the manuscript and we are certain this has helped improve its quality. Our responses to the review comments are below, and changes that we have made to the paper are highlighted in the revised manuscript submission.
Reviewer #3:
This paper presents an experimental study of LWC beams reinforced with FRP bars. The topic is within scope of this journal and may be interested by the readers. The authors may consider the following comments:
- Lines 74-81: The authors are asked to introduce that there are also SRPs, generlamente they are used as external reinforcement of structural elements. We segger to consider the work: https://doi.org/10.1016/j.engstruct.2022.114084.
|
Response: |
Reference is added |
- Line 140: What standard was used to evaluate the parameters shown in Table 3?.
|
Response: |
Different standards were used to evaluate the properties. As common in the papers in this area no need for adding every single reference for testing the properties of the materials used |
- Figure 3 is poorly readable.
|
Response: |
The resolution of the paper enhanced by zooming - |
- Lines 216-219: I am a little confused, was the test conducted in displacement control with a rate 1 mm/min or in load control "apply a monotonic increasing load"? The authors are invited to rewrite the sentence.
|
Response: |
The “displacement control” was with a rate of 1mm/min |
- Line 393: The authors are invited to expand the conclusions.
|
Response: |
Little modifications have applied |
Round 2
Reviewer 1 Report
The authors have addressed the comments properly.
Author Response
Thank you very much